# L-Dopa-Decarboxylase (DDC) Is a Positive Prognosticator for Breast Cancer Patients and Epinephrine Regulates Breast Cancer Cell (MCF7 and T47D) Growth In Vitro According to Their Different Expression of *G_i_- Protein- Coupled Receptors*

**DOI:** 10.3390/ijms21249565

**Published:** 2020-12-15

**Authors:** Eileen Tremmel, Christina Kuhn, Till Kaltofen, Theresa Vilsmaier, Doris Mayr, Sven Mahner, Nina Ditsch, Udo Jeschke, Aurelia Vattai

**Affiliations:** 1Department of Obstetrics and Gynecology, University Hospital, LMU Munich, Marchioninistraße 15, 81377 Munich, Germany; E.Tremmel@campus.lmu.de (E.T.); Till.Kaltofen@med.uni-muenchen.de (T.K.); Theresa.Vilsmaier@med.uni-muenchen.de (T.V.); Sven.Mahner@med.uni-muenchen.de (S.M.); Aurelia.Vattai@med.uni-muenchen.de (A.V.); 2Department of Obstetrics and Gynaecology, University Hospital Augsburg, Stenglinstr. 2, 86156 Augsburg, Germany; Christina.Kuhn@med.uni-muenchen.de (C.K.); nina.ditsch@uk-augsburg.de (N.D.); 3Department of Pathology, LMU Munich, Thalkirchner Straße 36, 80337 Munich, Germany; doris.mayr@med.uni-muenchen.de

**Keywords:** breast cancer, L-dopa decarboxylase, epinephrine

## Abstract

A coherence between thyroid dysfunction and breast cancer incidence exists. Thyroid hormone metabolites bind to TAAR1 (trace amine-associated receptor 1) and through that modulate the serotonergic and dopaminergic system. Catecholamines themselves are synthesized by the L-dopa decarboxylase (DDC). The aim of our study was to analyze the influence of catecholamines on the DDC expression in primary breast cancer patients and the role of DDC concerning overall survival (OS). DDC expression was analyzed by immunohistochemistry. The effect of epinephrine on the expression of DDC and the $G_{i}$- protein was analyzed on the protein level via Western blot. A viability assay was performed to test the metabolic cell viability. The overexpression of DDC in the primary tumor was associated with longer OS (*p* = 0.03). Stimulation with epinephrine induced the downregulation of DDC (*p* = 0.038) and significantly increased viability in T47D cells (*p* = 0.028). In contrast, epinephrine induced an upregulation of DDC and decreased the proliferation of MCF7 cells (*p* = 0.028). Epinephrine led to an upregulation of $G_{i}$ protein expression in MCF7 cells (*p* = 0.008). DDC is a positive prognostic factor for OS in breast cancer patients, and it is regulated through epinephrine differently in MCF7 and T47D. DDC may represent a novel target for the treatment of breast cancer, especially concerning its interaction with epinephrine.

## 1. Introduction

Breast cancer is the most common malignant cancer type in women worldwide [1]. In 2018, over two million patients were newly diagnosed with breast cancer, and more than 600,000 women died from it [2,3].

Various comorbidities can influence the biology and development of breast cancer. A correlation between breast cancer and thyroid disorders could be found [4]. The incidence of breast cancer in women with thyroid dysfunction is higher in comparison to normothyroidal women [4,5,6].

Sogaard et al. (2016) found a coherence between hyperthyroidism and an increased breast cancer incidence as well as a coherence between hypothyroidism and a lower breast cancer incidence [7]. Furthermore, Ditsch et al. (2017) described significantly elevated blood levels of thyroid hormones (fT_3_ and fT_4_)_,_ and high concentrations of TSH and thyroidal antibodies in breast cancer patients at the moment of primary diagnosis [8]. Thyroid hormones are metabolized by the enzyme ornithine decarboxylase (ODC) into the derivates called trace-amines (TAs) [9,10].

In recently published studies, investigations were made on the degradation products of thyroid hormones can bind to the trace amine-associated receptor 1 (TAAR1), which is a G-protein coupled receptor that influences the viability and migration of breast cancer cells [11,12,13,14]. Additionally, this receptor has been identified as an independent prognostic positive factor for breast cancer [15].

Further an enzyme called, L-dopa dopa decarboxylase (DDC), is a pyridoxal 5-phosphate (PLP)-dependent enzyme that converts L-dopa into dopamine, which is expressed ubiquitarily and similar to the ODC concerning its operation mode [16,17,18]. Additionally, DDC influences the synthesis of serotonin, tryptamine, phenylethylamine, and histamine [19]. In a former study, we could show that TAs have an influence on the expression of DDC in healthy pregnancies [19].

The degradation process of TAs is similar to the degradation and the synthesis of dopamine and serotonin [10,20]. Dopamine itself induces the synthesis of amines such as epinephrine and norepinephrine. DDC can influence the levels of catecholamines, epinephrine, and norepinephrine [21]. The enzyme DDC so far was investigated especially in neurological and psychiatric diseases due to its neurotransmitter building character [21]. DDC may act as a genetic modifier of the response to neurological treatment with L-dopa during Parkinson′s disease [22]. For different tumors such as neuroendocrine tumors, neuroblastoma, or prostate cancer, DDC is regarded as a biomarker [17]. High DDC expression has been observed in small-cell lung carcinoma (SCLC), neuroblastoma, pheochromocytoma, and in peritoneal dissemination of gastric carcinoma [17]. DDC is also known as a positive prognostic factor in colorectal adenocarcinoma, where the overall survival rate is higher in patients with a higher level of DDC expression [17].

DDC further influences the expression of dopaminergic D2 receptor (D2R) [23]. D2R is especially examined in neurological diseases such as Parkinson’s disease, dementia, or depression [24].

Epinephrine is a ligand that binds to D2R. It is an amine induced by dopamine, which is synthesized by the DDC. A defect of DDC causes a minor amount of catecholamines and neurotransmitters, such as dopamine and hence a minor amount of epinephrine [25,26]. Epinephrine is a physiological transmitter of the sympathetic nerve system [27]. This transmitter can bind to adrenoreceptors [28]. DDC is co-expressed with G-protein coupled receptors such as α and β adrenoreceptors (AR) in tumor cells [18].

Lüthy et al. (2009) described the function of α- and β-AR in breast structures and breast cancer, as well as their occurrence in other cancer identities [29]. Increased epinephrine levels can inhibit apoptosis in prostate cancer and stimulate tumor growth through the cAMP pathway in mice [30].

The binding of epinephrine onto the G-protein coupled ARs can stimulate or inhibit the adenylcylase (PKA) depending on the G-protein, which is bound. PKA influences the cAMP levels of the cell, which regulate the hormones syntheses of steroids or estrogen on a nuclear level through a cAMP-responsive element binding protein (CREB) [31,32].

The aim of our study was to analyze the influence of DDC expression in breast cancer cells in vivo and in vitro and to specify the interaction of G-protein coupled receptors with epinephrine. Additionally, we further investigated the signaling pathway of the DDC for the viability of breast cancer cells.

## 2. Results

### 2.1. DDC Protein Expression in Primary Breast Cancer

Via immunohistochemical analysis of 235 samples of breast cancer, a significant upregulation of DDC protein expression could be observed in primary breast cancer with a tumor size below pT2 (r = -0.147; *p* = 0.032) (Figure 1). Additionally, an upregulation of DDC expression could be analyzed via immunochemistry with higher tumor grading (G1–G2 *p* = 0.013; G1–G3 *p* = 0.008)) (Figure 2). An overexpression of DDC protein expression (intensity ≥ 2, *n* = 109) in the primary tumor was associated with a significantly longer overall survival (*p* = 0.03) of patients in the cohort compared to patients without a DDC overexpression (*n* = 126) dependent on the size of the tumor (Figure 3). A high expression of DDC is correlated with the tumor size and is a positive factor for the overall survival of breast cancer patients. Additionally, we detected a negative correlation with nuclear TAAR1 (cc = -0.175; *p* = 0.010) (Appendix A). No correlation could be observed between DDC expression in primary breast carcinoma and the histological subtypes, luminal A or B classification, progesterone receptor status (PR), estrogen receptor status (ER), HER-2 status, HER-4 status, age, or lymph node status.

### 2.2. Influence of Epinephrine on DDC Protein Expression in MCF7 and T47D Breast Cancer Model Cells

The incubation of T47D cells with 10 μM epinephrine induced a significant downregulation of DDC protein expression (*p* = 0.038) in our Western blot analysis (Figure 4). In MCF7 cells, an upregulation trend in DDC protein expression could be observed (*p* = 0.051) that is not significant through the incubation with 10 μM epinephrine after 6 h (Figure 4).

### 2.3. Viability of MCF7 and T47D Cells after Stimulation with Epinephrine

WST-1 Assay. Stimulation of both cell lines (MCF7, T47D) with 1 μM epinephrine for 48 h led to a significantly decreased proliferation of MCF7 cells (*p* = 0.028) in comparison to unstimulated MCF7 control cells (Figure 5). In contrast, T47D cells showed a significantly increased viability (*p* = 0.028) after stimulation with epinephrine in comparison to unstimulated T47D control cells (Figure 5).

### 2.4. Influence of Epinephrine on $G_{i}$- Protein -Expression in MCF7 and T47D Breast Cancer Model Cells

Incubation of T47D cells with 10 μM epinephrine for 6 h induced no significant difference in $G_{i}$- protein expression (*p* = 0.859). In contrast, MCF7 cells showed a significant change in $G_{i}$- protein expression. After an incubation with 10 μM epinephrine for 6 h, a significant upregulation of $G_{i}$- protein expression was observed (*p* = 0.008) (Figure 6). Furthermore, unstimulated MCF7 and T47D cells differed in their total expression of $G_{i}$- protein. MCF7 unstimulated cells expressed significantly less $G_{i}$- protein than unstimulated T47D cells (*p* = 0.008) (Figure 7). A trend of enhanced $G_{i}$- protein expression (*p* = 0.051) could be observed between MCF7 and T47D stimulated cells (Figure 7).

## 3. Discussion

In this study, DDC expression has been analyzed in vivo and in vitro in MCF7 and T47D breast cancer cells. Additionally, the influence of epinephrine on the DDC expression has been analyzed. We focused on the signaling pathway of the $G_{i}$- protein coupled receptor ($G_{i}\;$-PcR) of the adrenergic system to analyze the effect of epinephrine in breast cancer cells. In a previous study, we investigated that TAAR1 overexpression (IRS ≥ 6) is a positive prognostic marker for the OS in early breast cancer. TAAR1 is regulated by trace amines (Tas) [14]. The synthesis of TAs and catecholamines such as epinephrine is processed by similar enzymes, ODC and DDC. The latter is a positive prognostic factor for the overall survival of breast cancer patients. This prognostic factor in vivo is regulated through stimulation with epinephrine, which we could analyze via Western blot analysis. Zhang et al. (2018) already described the influence of TAs on TAAR1 and the dopaminergic release in neuronal cells. The different effect on the different locations of TAAR1 (intra- and extracellular) is explained through the specific interaction and location of the $D_{2}$ receptors. In cells stimulated with TAs, L-Dopa seems to be upregulated through the signaling pathway via TAAR1, increasing cAMP and PKA. There is a negative correlation between DDC in cytoplasm, which metabolizes L-Dopa and nuclear TAAR1 through a feedback loop or shuttling process [33,34].

We detected a difference in the two investigated cell lines MCF7 and T47D on their reaction on epinephrine stimulation. Radde et al. (2014) already described bioenergetic differences for MCF7 and T47D by their estrogen receptor (ER) positivity concerning their behavior under stimulation with estrogen and tamoxifen [35]. According to Radde et al., the cell lines differ in their oxygen consumption rate, ATP level, proton leak, and the relation between mitochondrial and nuclear DNA. Furthermore, the reaction on 4-hydroxytamoxifen, an ER targeted therapy, diverged from increasing or decreasing the tolerance in oxygenic stress [35]. Both cell lines vary in their expression of progesterone (PR) and ER. MCF7 cells express a higher level of ER than PR, whereas T47D cells express a higher level of PR [36]. Additionally, the cells differ in their amount of $G_{i}$-PcR, as our results showed. This finding could be seen in a context with other G-PcRs such as the G-PcR 81. This receptor modulates the biology and microenvironment of breast cancer and was identified as a tumor-promoting receptor in breast cancer progression by Lee et al [37].

In our study, the cell lines showed different results of DDC expression on protein level and different viability levels after stimulation with epinephrine. Brandie et al. and Barzegar et al. already showed in their investigation that MCF7 and T47D cell lines differ in various bioenergetic or apoptosis characteristics [35,38]. In accordance with their results, we could show that MCF7 and T47D as well differ in their viability depending on the stimulus of epinephrine. The exact interesting pathway stimulated by epinephrin is part of further investigations. So far, Jahanafrooz et al. showed similar interesting results with a natural polyphenol, Silibinin, which caused a cell cycle arrest that was equal with the viability in MCF7 cells but not in T47D cells [39]. Tremmel et al. showed in their latest study that MCF7 and T47D cells show different effect after T1AM stimulation on TAAR1 expression and that T47D cells seem to need estrogen supplementation in this case. [14] We further focused on $G_{i}$-PcR, which interacts with epinephrine [40]. Our results showed a higher reaction of MCF7 cells after stimulation with epinephrine, as the $G_{i}$-PcR expression after stimulation with epinephrine increased significantly in MCF7 cells.

Additionally, DDC protein expression in MCF7 cells was increased compared to T47D cell expression after stimulation with epinephrine.

Earlier studies showed that the viability of MCF7 cells is sensitive on estrogen level, whereas T47D cells are estrogen resistant [14,41]. The estrogen level is regulated through $G_{i}$-PcR similar to the α- AR, which is expressed in breast cancer cells [29].

Meta-analysis already showed a better clinical outcome depending on the AR status for breast carcinoma patients [42]. In ER-positive breast cancer cells, AR confers to be a good prognostic factor [43]. In ER-negative cases, the involvement of AR in promoting cell proliferation has also been observed [44].

Through stimulation with epinephrine, which acts via ARs, MCF7 cell viability and metabolism was reduced. MCF7 cell viability and metabolism is estrogen dependent [45]. The level of estrogen seems to drop under stimulation with epinephrine. Less estrogen leads to a reduction of the MCF7 cell viability and metabolism. This finding is in line with the investigation of O’ Mahony et al. (2012), who focused on the impact of estradiol on the metabolism of glycolysis of MCF7 cells and showed that in a setting of low extracellular glucose levels, *17-estradiol* rescues cell viability through the upregulation of ATP and the activity of pyruvate dehydrogenase, which induces the tricarboxylic acid cycle and suppresses glycolysis [46].

Ouyang et al. showed an increased malignancy in breast cancer cells after chronic stress, which indicates that there is a difference between the chronic and acute effect of epinephrine and a difference in vivo and in vitro [47]. Furthermore, the viability of MCF7 cells compared with the viability of T47D cells seems to be more influenced by the epinephrine stimulation. Epinephrine acts through the $G_{i}$-PcR. Both cell lines have a different sensitivity for the stimulation with epinephrine, which is in accordance with the different expression levels of $G_{i}$-PcR after stimulation. Steroids, such as estrogen or catecholamines, regulate the growth and development of the mammary gland as well as the development of breast neoplasms [41]. Stimulation with estrogen induces an in vitro proliferation of hormone-dependent breast cancer cells [48,49]. Epinephrine is able to downregulate the steroid and estrogen syntheses [50]. This is in line with our research results, indicating that epinephrine seems to induce a downregulation of estrogen synthesis and an upregulation of $G_{i}$-PcR in MCF7 cells. This signaling pathway should be part of further investigations. The signaling latter known so far involves the cAMP-nuclear binding element (CREB), which is regulated through the cAMP level [33]. cAMP is a product of the adenylate cyclase, which is inhibited through $G_{i}$-PcR. Aims of future studies include further investigation of the functional interaction between $G_{i}$-PcR and the DDC in breast cancer tissue and cells. Analyses of the signaling pathways of the endogenous ligands and a possible re-examination of adrenergic signaling in breast cancer need to be done.

## 4. Materials and Methods

### 4.1. Patients

Breast cancer samples were collected from 235 patients, who underwent surgery due to a malignant breast tumor at the Department of Gynaecology and Obstetrics of the Ludwig-Maximilians University hospital, Munich, Germany. Patient data were taken from patient charts or the Munich Cancer Registry. No positive family history nor metastases were recorded in any of those cases. Over 50% of the patients were diagnosed with a tumor size smaller than 2 cm (n(pT1) = 153 (68%); n(pT2) = 66 (29.3%); n(pT3) = 1 (0.4%); n(pT4) = 5 (2.2%)). No lymph node metastases at the point of diagnosis were recorded in 56.7% of the patients. Further grading, Her2 status, estrogen receptor (ER), progesterone receptor (PR), and luminal classification was performed on the samples (Table 1). The mean age was 58.2 ± 13.3 years. The mean survival was 12.2 years (95% CI: 11.6–12.8 years), and 49 deaths were documented, while a mean follow-up of 9.8 years was reached.

### 4.2. Ethical Approval and Informed Consent

All procedures involving human participants were in accordance with the ethical standards of the institutional and/or national research committee and with the Helsinki declaration of 1964 and its later amendments or comparable ethical standards. The study was approved by the local ethics committee of the Ludwig-Maximilians University of Munich (reference number of approvals: 337-06, approval date: 26 January 2010).

### 4.3. Immunohistochemistry

Tissue samples were fixed in 3.7% formalin and embedded with paraffin. Formalin-fixed paraffin-embedded sections (3 µm) were deparaffinized in Roticlear (Carl Roth, Karlsruhe, Germany), rehydrated, and blocked with 6% H_2_O_2_/methanol for 20 min to deactivate the endogenous peroxidase. For rehydration of the slides, an alcohol gradient up to distilled water was used. In sodium citrate (pH = 6.0), the slides were cooked under pressure. A blocking solution was used to prevent the primary antibody from binding unspecifically (Polymer Kit, Zytomed Systems, Berlin, Germany). After preparing the tissue sections, they were incubated with the primary DDC antibody for 16 h at 4 °C (Table 2). Reactivity was detected by using the ZytoChem Plus HRP Polymer System (mouse/rabbit) (Zytomed, catalog-ID: POLHRP-100, Source BioScience, Nottingham, United Kingdom) in accordance to the manufacturer′s protocol, and staining and counterstaining were carried out with DAB and hemalum, respectively. Appropriate positive controls (placenta tissue) were included in each experiment. Each slide was evaluated with the semi-quantitative immunoreactive score (IRS) using a Leitz Diaplan microscope (Leitz, Wetzlar, Germany). The IRS calculation works by multiplying the intensity of cell staining (0: none; 1: weak; 2: moderate; 3: strong) with the percentage of positively stained cells (0: no staining; 1: <10% of the cells; 2: 11–50%; 3: 51–80%; 4: >80%).

### 4.4. Cell Culturing and Cell Stimulation

The cell lines MCF7 and T47D were used as models for the ductal breast carcinoma. The cells were cultured in DMEM (3.7 g/L NaHCO3, 4.5 g/L D-glucose, 1.028 g/L stable glutamine, and Na-Pyruvate; Biochrom, Berlin, Germany). After adding 10% heat-inactivated fetal calf serum (FCS; Biochrom) to the medium the solution was incubated at an atmospheric CO_2_ concentration of 5% and at 37 °C in an incubator to lyse.

MCF7 and T47D cells were separately grown in sterile 12 multi-well slides at a density of 500,000 cells/mL DMEM with 10% FCS. The medium was changed after 4 h to pure DMEM without any FCS. After 20 h, the cells were stimulated with 1 μM epinephrine (Sigma-Aldrich Chemie GmbH, Munich, Germany). The cells were stimulated for 6 h with 10 μM epinephrine in the case of Western blot samples. Control cells were incubated in parallel without any stimulants.

### 4.5. Western Blot

Stimulated cells were lysed for 30 min at 4 °C with 200 µL buffer solution consisting of a 1:100 dilution of protease inhibitor (Sigma-Aldrich, St. Louis, MO, USA) in RIPA buffer (Radioimmunoprecipitation assay buffer, Sigma-Aldrich). Afterwards, the lysates were centrifugated, and a Bradford protein assay of the supernatant was performed. During Western blot, the proteins were separated according to their molecular weight using SDS-PAGE and transferred onto a PVDF membrane (Merck Millipore, Darmstadt, Germany). The membrane was blocked for 1 h in a receptacle containing Marvel-TBST for DDC-staining or containing of 1x Casein Solution (Vector Laboratories, Burlingame, CA, USA) for β-actin and $G_{i}$- staining (a Gα- protein inhibitor staining), against unspecific binding. The primary antibody Anti-DDC (polyclonal Rabbit IgG, LSBio) was diluted in a 1:1000 in Marvel-TBST (TBS + TWEEN and milk powder). The antibody Anti-ß-Actin (Clone AC-15, Mouse IgG, Sigma) was diluted in 1:1000, the antibody Anti- (polyclonal Rabbit IgG, Novus biological, Littleton, CO, USA) was diluted in 1:500 in casein, each according to the suggestion of the producer. The antibody Anti-DDC was applied for 1 h at room temperature. The antibodies for Anti-ß-Actin and Anti-$G_{i}$- were applied for 16 h at 4 °C. After rinsing with TBST, the membrane was incubated with biotinylated Anti-Rabbit IgG antibody and ABC-AmP reagent (both VECTASTAIN ABC-AmP Kit for rabbit IgG, Vector Laboratories) for DDC, ß-Actin, and $G_{i}$- Protein following the manufacturer′s protocol. BCIP/NBT chromogenic substrate (Vectastain ABC-AmP Kit, Vector Laboratories) showed specific bands on the membrane. Detection was performed with Bio-Rad Universal Hood II (Bio-Rad Laboratories, Hercules, CA, USA), and the bands were quantified using Bio-Rad Quantity One software (Bio-Rad Laboratories). Each Western blot experiment was validated nine times (*n* = 9)

### 4.6. WST-1 Assay

MCF7 and T47D cells were separately grown on sterile 98 multi-well slides at a density of 10.000 cells/mL DMEM containing 10% FCS. The medium was changed after 4 h to pure DMEM. After 16 h, cells were stimulated for 48 h with 1 μM epinephrine. The stimulants were renewed every 12 h. Control groups were incubated in pure DMEM. After stimulation, the reagent WST-1 water soluble tetrazolium (4-[3-(4-Iodophenyl)-2-(4-nitrophenyl)-2H-5-tetrazolio]-1,3-Benzol-Disulfonat) (Sigma-Aldrich, St. Louis, MO, USA) was applied in order to specify the mitochondrial succinate-tetrazolium dehydrogenase system of the stimulated cells. After 30 min of incubation with WST-1 in the incubator, cell viability was measured with a multi-well spectrophotometer at a wavelength of 420–480 nm. Each WST-1 Assay was validated nine times (*n* = 9).

### 4.7. Statistics

Data collection, processing, and analysis of statistical data were transacted with IBM SPSS Statistics for Windows, Version 22.0. Armonk, NY: IBM Corp. A significance level two-sided of 5% was used for all statistical tests. Kaplan–Meier curve analyses was performed to detect the overall survival rate. Non-parametric Mann–Whitney U test, Kruskal–Wallis one-way analysis of variance, or a two-sample t-test were performed to compare the central tendency. A Wilcoxon signed-rank test was used for statistical analysis of cell culture experiments. For DDC immunohistochemistry, an immunoreactive score (IRS) was used for each stained sample. The score obtains the values of the counted most perceived cells (IS = color intensity) multiplied by the positive cells (PP = percentage points).

## 5. Conclusions

L-dopa-decarboxylase could be identified as one positive prognosticator for breast cancer patients with primary tumors. Epinephrine, as a substrate of the DDC, influences the viability of breast cancer cells in vitro. DDC may represent a novel target for the prevention and treatment of breast cancer.

## Figures and Tables

**Figure 1 ijms-21-09565-f001:**
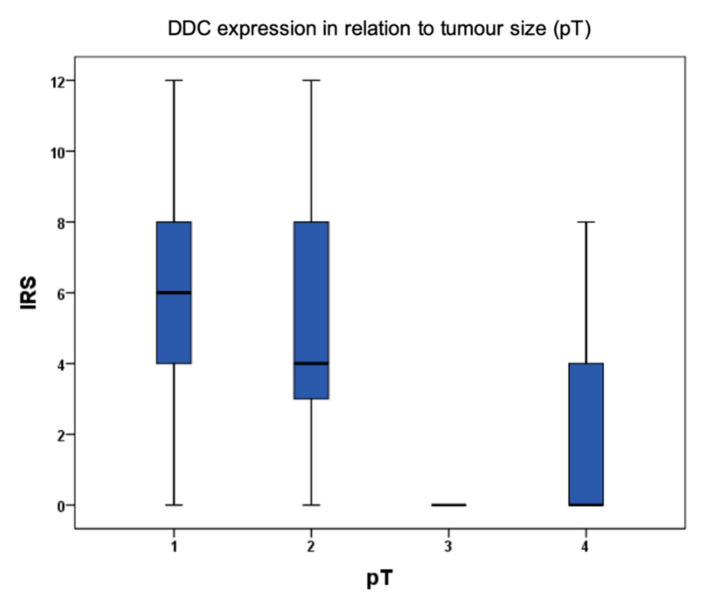
Dopa decarboxylase (DDC) protein expression in breast cancer cells according to tumor size. Boxplot of DDC protein expression in breast cancer in relation to tumor size and correspondent immunoreactive score (IRS) (r = −0.147; *p* = 0.032).

**Figure 2 ijms-21-09565-f002:**
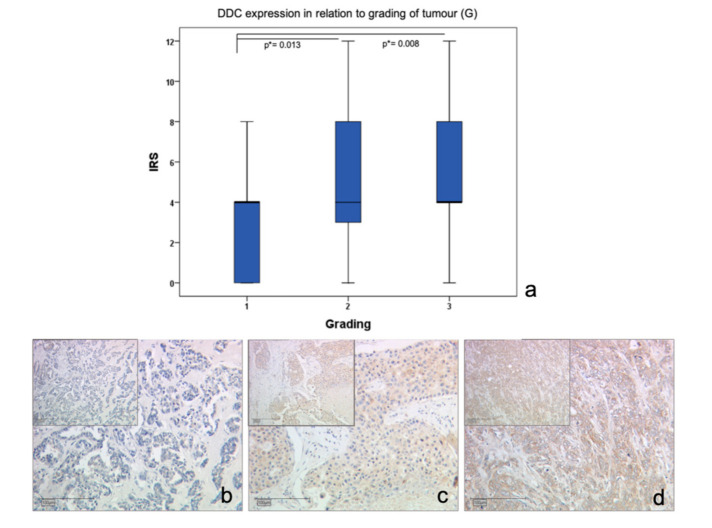
DDC protein expression in breast cancer cells according to tumor grading. (**a**) Boxplot of DDC protein expression in breast cancer according to their grading (* significant *p*-value), (**b**) DDC staining in breast cancer (grading 1) with IRS = 2, 10× and 25× magnification, (**c**) DDC staining in breast cancer (G2) with IRS = 4, 10× and 25× magnification, (**d**) DDC staining in breast cancer (G3) with IRS = 6, 10× and 25× magnification.

**Figure 3 ijms-21-09565-f003:**
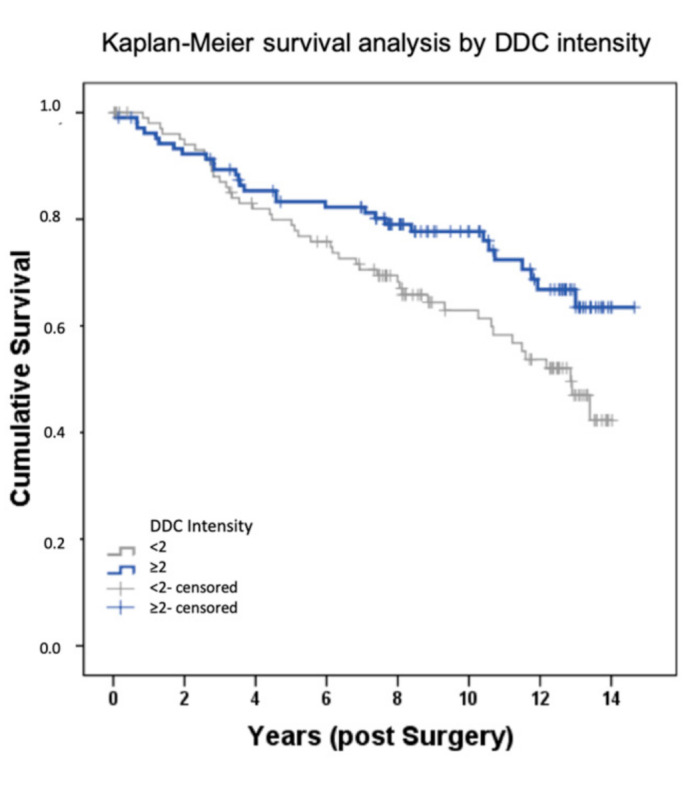
Influence of DDC expression on overall survival (OS) of patients with early breast cancer. Kaplan–Meier curve showing the association between an increased DDC expression (Intensity ≥ 2, *n* = 109; Intensity < 2, *n* = 126) and a longer overall survival (*p* = 0.03).

**Figure 4 ijms-21-09565-f004:**
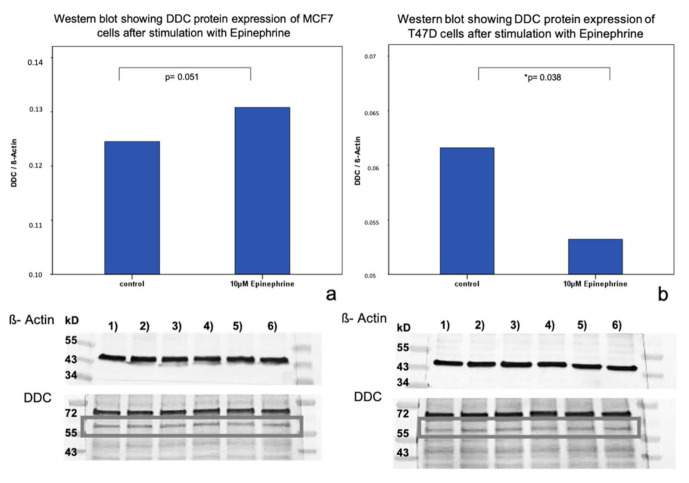
Western Blot analysis of DDC protein expression in MCF7 and T47D cells after stimulation with epinephrine. (**a**) Bar chart of DDC expression in MCF7 cells after incubation with 10 µM epinephrine. Epinephrine led to an increase of DDC protein expression (*p* = 0.051) that was not significant. 1)–6) Western blot membranes after incubation with ß-actin and DDC antibodies (marked with gray box). 1) 3) 5): control samples. 2) 4) 6): stimulated samples with 10 µM epinephrine. (**b**) Bar chart of DDC expression in T47D cells after incubation with 10 µM epinephrine (* significant *p*-value). Epinephrine induced a significant downregulation of DDC protein expression (*p* = 0.038). 1)–6) show Western blot membranes after incubation with ß-actin and DDC antibodies (marked with a gray box). 1) 3) 5): control samples. 2) 4) 6): stimulated samples with 10 µM epinephrine.

**Figure 5 ijms-21-09565-f005:**
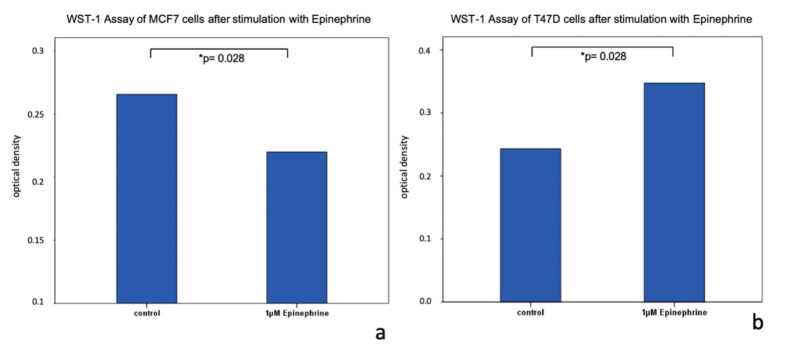
WST-1 assay of MCF7 and T47D cells stimulated with epinephrine. (**a**) Bar chart of optical density of MCF7 cells after incubation with 1 µM epinephrine for 48 h. Epinephrine induced a decrease of cell proliferation (*p* = 0.028; * significant *p*-value). (**b**) Bar chart of optical density of T47D cells after incubation with 1 µM epinephrine for 48 h. Epinephrine induced an increase of cell proliferation (*p* = 0.028; * significant *p*-value).

**Figure 6 ijms-21-09565-f006:**
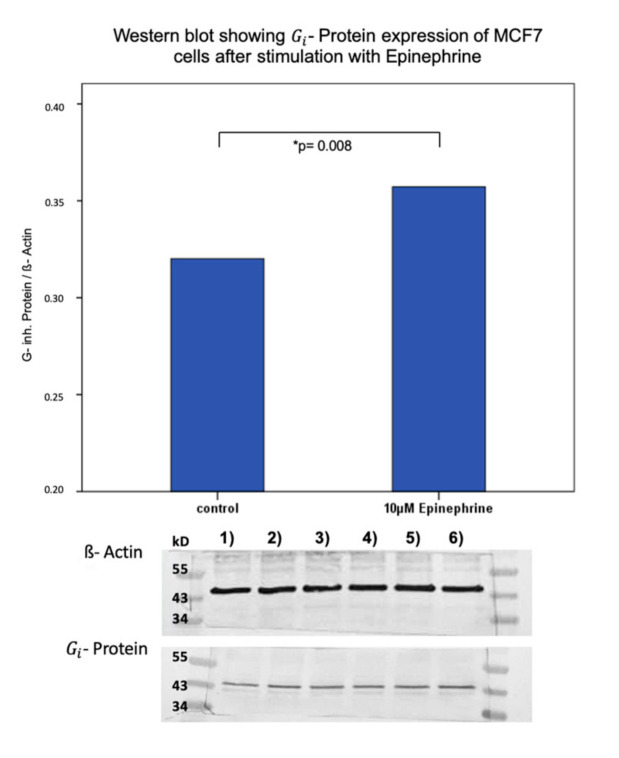
Western blot analysis of
$G_{i}$
- protein expression in MCF7 cells after stimulation with epinephrine. Bar chart of $G_{i}$- protein expression in MCF7 cells after incubation with 10 µM epinephrine. Epinephrine induced a significant upregulation of $G_{i}$- protein expression (*p* = 0.008; * significant *p*-value). 1)–6) Western blot membranes after incubation with ß-actin and $G_{i}$- protein antibodies. 1) 3) 5): control samples. 2) 4) 6): stimulated samples with 10 µM epinephrine (Appendix A).

**Figure 7 ijms-21-09565-f007:**
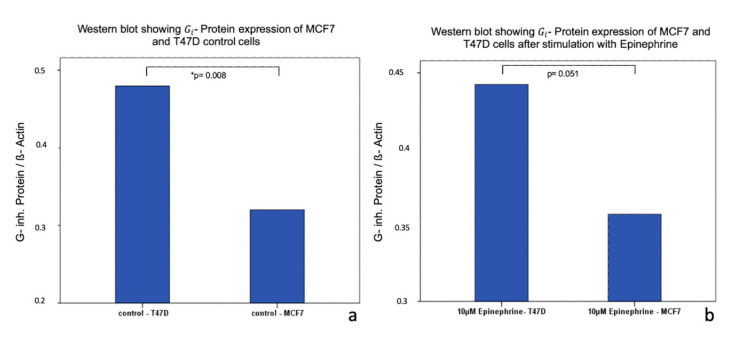
Western blot analysis of $G_{i}$- protein expression in MCF7 and T47D cells after stimulation with epinephrine. (**a**) Bar chart of $G_{i}$- protein expression in T47D and MCF7 control cells. There is a significant lower of expression of $G_{i}$- protein (*p* = 0.008; * significant *p*-value). (**b**) Bar chart of $G_{i}$- protein expression in MCF7 and T47D cells after incubation with 10 µM epinephrine have a tendency for a higher expression of $G_{i}$- protein in T47D cells after stimulation than in MCF7 cells after stimulation with 10 µM epinephrine (*p* = 0.051).

**Table 1 ijms-21-09565-t001:** Patient’s characteristics.

Patient’s Characteristics		*n*=
*size*		
	pT1	135 (68%)
	pT2	66 (29.3%)
	pT3	1 (0.4%)
	pT4	5 (2.2%)
*grading*		
	G1	16
	G2	84
	G3	57
*receptor status*		
Her2	positive	26
	negative	208
ER	positive	192
	negative	43
PR	positive	141
	negative	94
*luminal*		
	A	103
	B	74
*mean age*		
	58.2 years ± 13.3	

**Table 2 ijms-21-09565-t002:** Features of the antibodies used for staining.

Antibody	Incubation	Blocking Solution	Blocking Condition
DDC, Polyclonal (Rabbit IgG, LSBio)	1:300 in PBS 16 h at 4°C	Reagent I (Polymer Kit, Zytomed Systems, Berlin, Germany)	5 min

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
