# Peer review of "L-Dopa-Decarboxylase (DDC) Is a Positive Prognosticator for Breast Cancer Patients and Epinephrine Regulates Breast Cancer Cell (MCF7 and T47D) Growth In Vitro According to Their Different Expression of Gi- Protein- Coupled Receptors"

_ijms, 2020, doi:10.3390/ijms21249565_

Round 1
Reviewer 1 Report
This work by Tremmel et al. focuses on the study of DCC as a prognostic marker in breast cancer, and how epinephrine regulates the in vitro proliferation of breast cancer cell lines.
The work does not provide robust data that allow conclusions to be drawn. Besides, the wording is not careful, both the abstract and the introduction can be improved, there is disjointed information and it is not contextualized in breast cancer. Also, the title becomes quite dense. No relevant conclusion is reached in the discussion. The results obtained are merely discussed and the objectives set are not justified.
Regarding the results
In all the graphs the error bars and knowing the n used for the statistical analysis are missing. I find it hard to believe that significant p values ​​are obtained with this variability. In Figure 1 the groups cannot be compared since T3 there is only 1 patient and in T4, 5.
TAAR1 (nucleus-cytoplasm) data is not documented
The densitometry data from WB is not very credible and error bars are missed. How they have been done. What statistic was used?
In the figure caption, it is mentioned T1AM instead of epinephrine, it should be explained.
In Materials and methods, a table with the clinical characteristics of the patients is missing
The section describing the statistical analyzes is incomplete
L188-189, not discussed with other results (for example, https://www.ncbi.nlm.nih.gov/pmc/articles/PMC5342597/)
The comparison that is made between the two cell lines is not understood, and what the title indicates is misplaced as both cell lines give opposite results in response to epinephrine.
From a formal point of view:
References must be indicated before the end of the sentence. In L52 a connector is missing so that the phrase is understood.
In the Kaplan-Meier indicate IRS <2; IRS ≥ 2 (instead of 2 or more)
L120-121, modify the phrase or join the two. It is said that it is not significant and then the opposite.
In the text, figure 7 is mentioned first before figure 6
L144-146, it doesn't make sense
Author Response
Responses to the reviewers
L-DOPA-Decarboxylase (DDC) is a positive prognosticator for breast cancer patients and Epinephrine regulates breast cancer cell growth in vitro according to DDC regulation
Reviewer 1
Comments and Suggestions for Authors
This work by Tremmel et al. focuses on the study of DCC as a prognostic marker in breast cancer, and how epinephrine regulates the in vitro proliferation of breast cancer cell lines.
The work does not provide robust data that allow conclusions to be drawn. Besides, the wording is not careful, both the abstract and the introduction can be improved, there is disjointed information, and it is not contextualized in breast cancer. Also, the title becomes quite dense. No relevant conclusion is reached in the discussion. The results obtained are merely discussed and the objectives set are not justified.
Thank you for your thoughtful and constructive annotation.
The paper summarizes our results, which build a basis for further studies. We agree with you that further investigations on basis of this paper are needed. The contextualization of breast cancer is given in the introduction section (line 37- 68). We introduce our study and publications about former studies of our research group, and which served as a basis for the current study. Our study focuses on breast cancer biology with a new aspect of DDC expression and epinephrine. It is an important remark to compare the importance of the results with the results of studies other research groups and fully agree with you in this point.
The relevant conclusion that DDC is a positive prognostic factor in primary breast cancer is mentioned in the discussion section (line 202) and in the rewritten conclusion section (line 449).
We improved the Abstract section:
A coherence between thyroid-dysfunction and breast cancer incidence exists. Thyroid hormone metabolites bind to the TAAR1 and through that modulate the serotonergic and dopaminergic system. Catecholamines themselves are synthesized by the L-Dopa Decarboxylase (DDC). Aim of our study was to analyze the influence of catecholamines on the DDC expression in primary breast cancer patients and the role of DDC concerning overall survival (OS). DDC expression was analysed by immunohistochemistry. The effect of epinephrine on the expression of DDC and the - Protein was analysed on protein level via Western Blot. A WST-1 assay was performed to test the metabolic cell viability. Overexpression of DDC in the primary tumor was associated with longer OS (p=0.03). Stimulation with epinephrine induced downregulation of DDC (p=0.038) and significantly increased viability in T47D cells (p=0.028). In contrast, epinephrine induced an upregulation of DDC and decreased proliferation of MCF7 cells (p=0.028). Epinephrine led to an upregulation of - protein expression in MCF7 cells (p=0.008). DDC is a positive prognostic factor for OS in breast cancer patient and it is regulated through epinephrine differently in MCF7 and T47D. DDC may represent a novel target for the treatment of breast cancer especially concerning its interaction with epinephrine.
We rewrote the conclusion section (line 410):
L-DOPA-Decarboxylase could be identified as one positive prognosticator for breast cancer patients with primary tumor. Epinephrine, as a substrate of the DDC, influences the viability of breast cancer cells in vitro. DDC may represent a novel target for the prevention and treatment of breast cancer.
Regarding the results
- In all the graphs the error bars and knowing the n used for the statistical analysis are missing. I find it hard to believe that significant p values ​​are obtained with this variability.
Please excuse the error. We always used a Western Blot and WST study design n=9, so significant p- values could be detected. The experiments were validated nine times in case of Western Blot and WST (n=9).
The bar charts show the mean of relative expression and not the raw data. Therefore, the presentation of error bars is not applicable in this case. [1] It is not common to add error bars on mean values of relative expression.
We added the following paragraphs:
Line 387: Each Western Blot experiment was validated nine times (n= 9).
Line 398: Each WST-1 Assay was validated nine times (n=9).
- In Figure 1 the groups cannot be compared since T3 there is only 1 patient and in T4, 5.
Thank you for your remark. This is a relevant annotation. With Kruskal–Wallis one-way analysis of variance it is possible to make statements in this group population. The collective of patients is an accidental collective, which is an advantage. It is no pre adjustment collective. That is positive, because it gives a realistic presentation of the distribution of the breast cancer size in reality the best. This collective has been used for several publications; therefore, it builds a profound basis for further investigations.[2,3] The fact that breast cancer has been diagnosed in our collective in 50% of the patients with a tumor size smaller than 2cm (<T2) is in line with the reality in clinical setting. Breast cancer is rarely first diagnosed at stage T3 or T4. Therefore, in this study we focused on the total >T2 and <T2. Even if it is not equal, it does represent the breast cancer distribution in the population.
- TAAR1 (nucleus-cytoplasm) data is not documented
This data is based on the immunohistochemistry data of our laboratory; it showed a correlation with our study regarding TAAR1 which we quoted in the introduction. [4,5] Your annotation was really helpful, so we created a supplementary table (supplementary table 1) concerning the patients’ characteristics and staining profile.
It is definitely a result we further have to investigate on. TAAR1 is an intracellular amine-activated -coupled and -coupled receptor. [4] This data shows a negative correlation of nuclear TAAR1 staining and DDC staining in cytoplasm. It would be interesting to investigate further on a theory of shuttling between core and cytoplasm.[6] This needs to be proofed in future with further investigations.
We completed the discussion section (line 232):
There is a negative correlation between DDC in cytoplasm which metabolizes L-Dopa and TAAR1 in the core through a feedback loop or shuttling process. [33,34] This needs to be further analyzed.
- The densitometry data from WB is not very credible and error bars are missed. How they have been done. What statistic was used?
Thank you for your remark. We validated the Western Blot through densitometry nine times. Wilcoxon signed-rank test was used for statistical analysis of cell culture experiments like Western Blot, to evaluate paired groups. The bar charts show the mean of relative expression and not the raw data. Therefore, the presentation of error bars is not applicable and not common in this case. [1]
- In the figure caption, it is mentioned T1AM instead of epinephrine, it should be explained.
Thank you for your annotation, that was incorrect. We corrected the mistake in line 127 and line 137.
- In Materials and methods, a table with the clinical characteristics of the patients is missing
This is a very good annotation, thank you. The characteristics of the patients for the immunohistochemistry are summarized in Materials and Methods in wording (line 347- 356) and we added a table with patients’ characteristics (table 1).
We added table 1 with the characteristics of the collective in line 316:
- The section describing the statistical analyzes is incomplete
Thank you for this relevant comment. We should have mentioned the Kaplan- Meier analyses and should have commented on how we validated the immunohistochemical analysis. The statistic section includes now all statistical analyses we used according to the recommendation of our statistic masters at LMU Munich. Those statistical analyses have been used in several publications of our study group. [4,5,7]
We completed the paragraph (line 402):
Kaplan- Meier curve analysis was performed to detect the overall survival rate. Non-parametric Mann-Whitney U test, Kruskal–Wallis one-way analysis of variance or two sample t-test were performed to compare the central tendency. Wilcoxon signed-rank test was used for statistical analysis of cell culture experiments. For DDC immunohistochemistry an immunoreactive score (IRS) was used for each stained sample. The score obtains the values of the counted most perceived cells (IS= color intensity) multiplied by the positive cells (PP= percentage points).
- L188-189, not discussed with other results (for example, https://www.ncbi.nlm.nih.gov/pmc/articles/PMC5342597/)
Thank you for this important annotation.
We added the paragraph (line 243):
Additionally, the cells differ in their amount of - PcR as our results showed. This finding could be seen in a context with other G-protein-coupled receptors such as the G-protein- coupled receptor 81. This receptor modulates the biology and microenvironment of breast cancer and was identified as a tumor-promoting receptor in breast cancer progression by Lee et al.. [8]
- The comparison that is made between the two cell lines is not understood.
Different study groups pointed out that MCF7 and T47D differ in various characteristics. [4,9,10] That is why it is so interesting to compare those two cell lines. The aim is to identify and specify those and to investigate differential behavior of breast cancer cells. In order to show this variety, we completed the discussion section in order for the reader to understand the complexity of differences more easily.
We added the following paragraph (line 241):
Both cell lines vary in their expression of progesterone (PR) and estrogen receptors (ER). MCF7 cells express a higher level of ER than PR, whereas T47D cells express a higher level of PR. [36] Additionally, the cells differ in their amount of - PcR as our results showed. This finding could be seen in a context with other G-PcRs. Like the G-PcR 81, which modulates the biology and microenvironment of breast cancer and was identified as a tumor-promoting receptor in breast cancer progression by Lee et al.. [37] In our study, the cell lines showed different results of DDC expression on protein level and different viability levels after stimulation with epinephrine. Brandie et al. and Barzegar et al. already showed in their investigation, that MCF7 and T47D cell lines differ in various bioenergetic or apoptosis characteristics. [38,39] In accordance with their results we could show that MCF7 and T47D as well differ in their viability dependent of the stimulus in our case Epinephrine. The exact interesting pathway stimulated by epinephrin is part of further investigations. Hence, the metabolic cell activity of MCF7 cells seems to be reduced through epinephrine compared to T47D cells. So far, Jahanafrooz et al. showed similar interesting results with a natural polyphenol, Silibinin, that caused a cell cycle arrest, which is equal with the viability, in MCF-7 cells but not in T47D cells. [40] We further focused on -PcR, which interacts with epinephrine. [41] Our results showed a higher reaction of MCF7 cells after stimulation with epinephrine, as the - PcR expression after stimulation with epinephrine increased significantly in MCF7 cells.
- What the title indicates is misplaced as both cell lines give opposite results in response to epinephrine.
Thank you for your remark and careful reading. We want to be as clear as possible in the title, so we modulated it. We have improved it concerning the results of our study.
We corrected the title:
L-DOPA-Decarboxylase (DDC) is a positive prognosticator for breast cancer patients and Epinephrine regulates breast cancer cell (MCF7 and T47D) growth in vitro according to their different expression of - protein- coupled receptors
From a formal point of view:
- References must be indicated before the end of the sentence.
The references are indicated with der Endnote citations style MDPI, which is mentioned on the IJMS homepage(https://www.mdpi.com/journal/ijms/instructions). The references where indicated after the full stop by the Endnote program itself in this citation style. If the paper is going to be accepted, the IJMS editors for sure could change this circumstance easily if needed.
- In L52 a connector is missing so that the phrase is understood.
Thank you for your comment on this phrase.
We added the following paragraph (line 74) to clarify:
Further an enzyme called, DOPA-decarboxylase (DDC), is a pyridoxal 5-phosphate (PLP)-dependent enzyme, that converts L-dopa into dopamine, which is expressed ubiquitarily and which is similar to the ODC concerning its operation mode.
- In the Kaplan-Meier indicate IRS <2; IRS ≥ 2 (instead of 2 or more)
Thank you for this correction. Please excuse the mistake. We have corrected it accordingly.
- L120-121, modify the phrase or join the two. It is said that it is not significant and then the opposite.
This is a helpful annotation.
We modified the phrase (line 142):
In MCF7 cells, an upregulation trend in DDC protein expression could be observed (p= 0.051) which is not significant, through the incubation with 10mM epinephrine after 6h (Figure 4).
- In the text, figure 7 is mentioned first before figure 6
This is a very good correction, thank you. We moved the sentence from line 217 to line 180 and modified it further to clarify it:
A trend of enhanced - protein expression (p=0.051) could be observed between MCF7 and T47D stimulated cells (Figure 7).
- L144-146, it does not make sense
We corrected the abstract, the T1AM was incorrectly mentioned in this context. Thank you.
Figure 5 WST-1 Assay of MCF7 and T47D cells stimulated with Epinephrine. a) Bar chart of optical density of MCF7 cells after incubation with 1 µM Epinephrine for 6h. Epinephrine induced a decrease of cell proliferation (p= 0.028). b) Bar chart of optical density of T47D cells after incubation with 1 µM Epinephrine for 6h. Epinephrine induced an increase of cell proliferation (p=0.028).
References
- Kolben TM, R.E., Vattai A, Hester A, Kuhn C, Schmoeckel E, Mahner S, Jeschke U, . PPARγ Expression Is Diminished in Macrophages of Recurrent Miscarriage Placentas.
- Ditsch, N.; Liebhardt, S.; Von Koch, F.; Lenhard, M.; Vogeser, M.; Spitzweg, C.; Gallwas, J.; Toth, B. Thyroid function in breast cancer patients. Anticancer Res 2010, 30, 1713-1717.
- Schröder, L.; Marahrens, P.; Koch, J.G.; Heidegger, H.; Vilsmeier, T.; Phan-Brehm, T.; Hofmann, S.; Mahner, S.; Jeschke, U.; Richter, D.U. Effects of green tea, matcha tea and their components epigallocatechin gallate and quercetin on MCF‑7 and MDA-MB-231 breast carcinoma cells. Oncology reports 2019, 41, 387-396, doi:10.3892/or.2018.6789.
- Tremmel, E.; Hofmann, S.; Kuhn, C.; Heidegger, H.; Heublein, S.; Hermelink, K.; Wuerstlein, R.; Harbeck, N.; Mayr, D.; Mahner, S., et al. Thyronamine regulation of TAAR1 expression in breast cancer cells and investigation of its influence on viability and migration. Breast Cancer (Dove Med Press) 2019, 11, 87-97, doi:10.2147/bctt.S178721.
- Vattai, A.; Akyol, E.; Kuhn, C.; Hofmann, S.; Heidegger, H.; von Koch, F.; Hermelink, K.; Wuerstlein, R.; Harbeck, N.; Mayr, D., et al. Increased trace amine-associated receptor 1 (TAAR1) expression is associated with a positive survival rate in patients with breast cancer. J Cancer Res Clin Oncol 2017, 10.1007/s00432-017-2420-8, doi:10.1007/s00432-017-2420-8.
- Wang, Y.; Tu, K.; Liu, D.; Guo, L.; Chen, Y.; Li, Q.; Maiers, J.L.; Liu, Z.; Shah, V.H.; Dou, C., et al. p300 Acetyltransferase Is a Cytoplasm-to-Nucleus Shuttle for SMAD2/3 and TAZ Nuclear Transport in Transforming Growth Factor β-Stimulated Hepatic Stellate Cells. Hepatology (Baltimore, Md.) 2019, 70, 1409-1423, doi:10.1002/hep.30668.
- Gratz, M.J.; Stavrou, S.; Kuhn, C.; Hofmann, S.; Hermelink, K.; Heidegger, H.; Hutter, S.; Mayr, D.; Mahner, S.; Jeschke, U., et al. Dopamine synthesis and dopamine receptor expression are disturbed in recurrent miscarriages. Endocr Connect 2018, 7, 727-738, doi:10.1530/ec-18-0126.
- Lee, Y.J.; Shin, K.J.; Park, S.A.; Park, K.S.; Park, S.; Heo, K.; Seo, Y.K.; Noh, D.Y.; Ryu, S.H.; Suh, P.G. G-protein-coupled receptor 81 promotes a malignant phenotype in breast cancer through angiogenic factor secretion. Oncotarget 2016, 7, 70898-70911, doi:10.18632/oncotarget.12286.
- Radde, B.N.; Ivanova, M.M.; Mai, H.X.; Salabei, J.K.; Hill, B.G.; Klinge, C.M. Bioenergetic differences between MCF-7 and T47D breast cancer cells and their regulation by oestradiol and tamoxifen. The Biochemical journal 2015, 465, 49-61, doi:10.1042/bj20131608.
- Barzegar, E.; Fouladdel, S.; Movahhed, T.K.; Atashpour, S.; Ghahremani, M.H.; Ostad, S.N.; Azizi, E. Effects of berberine on proliferation, cell cycle distribution and apoptosis of human breast cancer T47D and MCF7 cell lines. Iranian journal of basic medical sciences 2015, 18, 334-342.

Reviewer 2 Report
The manuscript by Tremmel et al. investigates L-Dopa decarboxylase (DDC) expression in primary breast cancer and epinephrine-induced DDC and Gi-protein expression in MCF7 and T47D breast cancer cells. They concluded that DDC is a positive prognostic factor, and DDC may represent a novel target for breast cancer treatment. Their observation may be potentially meaningful in that DDC expression is elevated in primary breast cancer as revealed by immunohistochemistry, and its expression is associated with long survival of breast cancer patients as revealed by Kaplan-Meier survival analysis. However, the major shortcoming of this manuscript is that most of the data presented throughout Figures 4-7 appear to be preliminary. The molecular mechanisms underlying epinephrine-induced DDC and Gi-protein expression were unclear. I feel that it is overly conclusive to evaluate DDC as a novel target for the prevention and treatment of breast cancer without more detailed functional studies.
Major points
(1) Figure 4. Western blot data did not clearly show the difference between the control and epinephrine-treated groups. (i) The authors should present more sensitive experiments, such as RT-PCR or qPCR. (ii) Why were the cells treated for 6 h to detect DDC protein expression levels? Is DDC an immediate-early response gene? The authors should be required to conduct time-course experiments.
(2) Figure 5. (i) Which time is correct; 48 h (line 135) or 6 h (line 140)? (ii) Different concentrations were used for viability assay (1 μM) and western blotting (10 μM). Dose-dependent experiments are required for cell viability assay.
(3) Figure 6. Similar to Figsure 4. (i) Why were the cells treated for 6 h to detect Gi-protein expression levels? Is Gi-protein a known immediate-early response gene? The authors should be required to conduct time-course experiments. (ii) The authors should present more sensitive experiments, such as RT-PCR or qPCR.
(4) Figure 7. Please provide western blot data as like in Figure 6. To compare the levels of Gi-protein expression between MCF7 and T47D cells, they should be compared in the same blot.
(5) Discussion. (i) Line 191. Metabolic cell activity is not sufficient to account for the different expression of DDC between MCF7 and T47D cells. (ii) Many studies have demonstrated that epinephrine stimulates MCF7 proliferation. The authors should discuss why epinephrine reduces the cell viability of MCF7 in this study. (iii) More discussion is needed as to why epinephrine-induced DDC and Gi-protein expression differ between MCF7 and T47D cells.
Minor points
(1) line 26. PCR data were not presented in this study.
(2) line 208. Reference.
(3) line 266. 0.1 μM-treated data were not provided in this study.
(4) Treated time periods were described differently in the figure legends and methods; line 140, 6 h; line 293, 48 h.
Author Response
Responses to the reviewers
L-DOPA-Decarboxylase (DDC) is a positive prognosticator for breast cancer patients and Epinephrine regulates breast cancer cell growth in vitro according to DDC regulation
Reviewer 2
Comments and Suggestions for Authors
The manuscript by Tremmel et al. investigates L-Dopa decarboxylase (DDC) expression in primary breast cancer and epinephrine-induced DDC and Gi-protein expression in MCF7 and T47D breast cancer cells. They concluded that DDC is a positive prognostic factor, and DDC may represent a novel target for breast cancer treatment. Their observation may be potentially meaningful in that DDC expression is elevated in primary breast cancer as revealed by immunohistochemistry, and its expression is associated with long survival of breast cancer patients as revealed by Kaplan-Meier survival analysis. However, the major shortcoming of this manuscript is that most of the data presented throughout Figures 4-7 appear to be preliminary. The molecular mechanisms underlying epinephrine-induced DDC and Gi-protein expression were unclear. I feel that it is overly conclusive to evaluate DDC as a novel target for the prevention and treatment of breast cancer without more detailed functional studies.
Thank you for your constructive annotation. We totally agree that our study is a small step of this new possible therapy target and that further investigations are needed. The molecular mechanisms which you mentioned are our next aim to clarify and investigate on. Evangelos et al. already identified the DDC as a relevant part of the cascade of cancer cell cytotoxicity and therapeutic mechanism under therapy with docetaxel. [1,2] This supports the argument why it is so interesting to focus on DDC as a possible relevant new therapy target.
Major points
(1) Figure 4. Western blot data did not clearly show the difference between the control and epinephrine-treated groups.
(i) The authors should present more sensitive experiments, such as RT-PCR or qPCR.
à Thank you for your remark on the experiments. We agree with your evaluation about Western Blot and PCR. In this study the expression on protein level was of high importance and so we mainly focused on Western Blot experiments. We validated each of our experiments nine times. The densitometry showed significant results for the DDC protein and the - Protein expression (validated through ß-Actin). Experiments like RT-PCR are sensitive experiments for genetic regulation. PCR evaluates the gene expression which is much more influenced on different levels and through a various number of mechanisms. Further PCR is more error prone concerning its time kinetics and it is of importance to find the special time window to validate the right amount of RNA expression. We performed RT- PCR with our stimulated cells. We tried different time spots to validate through PCR but did not find a significant one. Here further investigations are needed and planned to find the perfect time kinetics for DDC expression on genetic level.
(ii) Why were the cells treated for 6 h to detect DDC protein expression levels? Is DDC an immediate-early response gene? The authors should be required to conduct time-course experiments.
à Thank you for this constructive thought. Only the significant results have been demonstrated graphically. For each experiment we have tested in advance a variety of different concentrations on protein level. The concentrations 1nM and 10nM have been tested. Further, we have been tested different conduct time- course experiments with 1h, 3h, 6h and 12h. The experiments have been performed in advance as a pre- study in our laboratory. Another publication from our study group (Tremmel et al., 2019) is based on the similar pre- study.[3] We have chosen a short stimulation time as epinephrine has a short half-time. [4] In addition, we did not add new stimulation constantly during the stimulation period. Hence, 6h achieved significant Western Blot results. Additionally, we used the same stimulation time for the - Protein experiment in order to achieve equal conditions. Additionally, the - Protein is a rapid acting protein, which encouraged us to use 6h as a stimulation time.
(2) Figure 5.
(i) Which time is correct; 48 h (line 135) or 6 h (line 140)?
à Thank you for this correction and careful reading of our paper. 48h is correct. We corrected the mistake in the figure legend in line 140.
(ii) Different concentrations were used for viability assay (1 μM) and western blotting (10 μM). Dose-dependent experiments are required for cell viability assay.
à You are right, in the figures only the significant results have been demonstrated graphically. Dose-dependent experiments have been performed in our laboratory group for viability assay and Western Blot. For each Western Blot experiment we have tested in advance different concentrations on protein level. The concentrations 1nM and 10nM have been tested. The experiments have been validated nine times. For the WST assay, different pre -studies have been made in our study group, which showed the best results for stimulation with 1 μM epinephrine. We based our study on the knowledge of our laboratory and former studies. [3,5-7] Because of the short period of time given for the revision, we were not able to perform new experiments. We surely keep your annotation in mind for our further investigations on this topic.
(3) Figure 6. Similar to Figure 4.
(i) Why were the cells treated for 6 h to detect Gi-protein expression levels? Is Gi-protein a known immediate-early response gene? The authors should be required to conduct time-course experiments.
à We used the same period of time as we identified for DDC, so we could be sure the same conditions prevailed. This stimulation time was chosen together with the short half-time of epinephrine and it’s character as a short term acting catecholamine. [4] Further the stimulation time matched with the - protein expression, because the - protein is a fast reacting protein. [8]
(ii) The authors should present more sensitive experiments, such as RT-PCR or qPCR.
à Thank you for your remark on the experiments. As mentioned on remark 1(i) above, we agree with your evaluation about Western Blot and PCR. In our case the expression on protein level was of high importance and so we mainly focused on our results of Western Blot experiments. We validated each of our experiments nine times. And even the differences are hard to see with bare eyes on the membrane, the densitometry showed significant results for the DDC protein and the - Protein expression on protein level (validated through ß-Actin expression). Experiments like RT-PCR are sensitive experiments for genetic regulation. The PCR evaluates the gene expression which is much more influenced on different levels and through different mechanisms. Further it is more error prone concerning its time kinetics. You have to find the special time window to validate the right amount of RNA expression. We performed RT- PCR with our stimulated cells. We tried a few different time spots to validate through PCR but did not find a significant one. Here further investigations are needed and planned to find the perfect time kinetics for - Protein expression on genetic level.
(4) Figure 7. Please provide western blot data as like in Figure 6. To compare the levels of Gi-protein expression between MCF7 and T47D cells, they should be compared in the same blot.
à Thank you for this constructive and relevant comment. In this paper we wanted to focus on the significant results and demonstrate those graphically. For clarification, we provided a supplementary figure 1 with the bar chart for T47D cells stimulated with Epinephrine and blotted with - Protein antibodies.
Figure legend:
Supplementary figure 1. Western Blot analysis of - protein expression in T47D cells after stimulation with Epinephrine. Bar chart of - protein expression in T47D cells after incubation with 10µM Epinephrine. Epinephrine showed no significant impact on T47D (p= 0.859).
(5) Discussion.
(i) Line 191. Metabolic cell activity is not sufficient to account for the different expression of DDC between MCF7 and T47D cells.
à Thank you for your remark here. The metabolic cell viability not the activity is regulated different between MCF7 and T47D cells after stimulation with epinephrin, as shown in our study. You are right, we have corrected this. The cell lines differ in a various number of characteristics as also shown by Brandie et al. and Barzegar et al. [9,10] Additionally Jahanafrooz et al. found that a natural polyphenol, Silibinin, caused a cell cycle arrest, which is equal with the viability, in MCF-7 but not in T47D cells. [11] Further,in our latest publication we investigated differences in these two cell lines concerning their reaction after stimulation with estrogen.[3] Due to this finding it is important to compare those two cell lines in order to get a better understanding of their characteristics.
We added the paragraph (line 248):
Brandie et al. and Barzegar et al. already showed in their investigation, that MCF7 and T47D cell lines differ in various bioenergetic or apoptosis characteristics. In accordance with their results, we could show that MCF7 and T47D differ in their viability depending on the stimulus of epinephrine. The exact interesting pathway stimulated by epinephrine is part of further investigations.
(ii) Many studies have demonstrated that epinephrine stimulates MCF7 proliferation. The authors should discuss why epinephrine reduces the cell viability of MCF7 in this study.
à This is one point which is needed to be investigated in further studies. We see a difference between our study and for example Cui et al. or Ouyang et al. in the acute stimulation with epinephrine in vitro in our experiments and the chronic effect of stress in vivo.[12,13] We stimulated the cells for a short period of time and had a look on the acute happenings on protein level. We made investigations in vitro, where we have detected different effects in MCF7 and T47D cells after acute short-term stimulation with epinephrine, which is needed to be further investigated.
We added the paragraph (line 285):
Ouyang et al. showed an increased malignancy in breast cancer cells after chronic stress, which indicates that there is a difference between the chronic and acute effect of epinephrine and a difference in vivo and in vitro.
(iii) More discussion is needed as to why epinephrine-induced DDC and Gi-protein expression differ between MCF7 and T47D cells.
à Thank you for this annotation. MCF7 and T47D are often used similarly in in vitro experiments. But as you mentioned, they differ in various characteristics. That is the reason why it is so interesting und important to use those two cell lines, to investigate more and get a better understanding on their signaling pathways. The DDC and Gi-protein expression difference is definitely one big topic, where additional investigations have to be carried out. Our study group is already planning further experiments. As already mentioned, the cell lines differ in various characteristics, shown by Brandie et al., Barzegar et al., Jahanafrooz et al. or Tremmel et al.. Those differences are not fully understood so far. We tried our best to explain our novel results and are planning to investigate further on this topic.
We added the paragraph (line 252):
So far, Jahanafrooz et al. showed similar interesting results with a natural polyphenol, Silibinin, that caused a cell cycle arrest, which is equal with the viability, in MCF-7 cells but not in T47D cells. [11] Tremmel et al. showed in their latest study that MCF7 and T47D cells show different effect after T1AM stimulation on TAAR1 expression and that T47D cells seem to need estrogen supplementation in this case. [14]
Minor points
(1) line 26. PCR data were not presented in this study.
à Thank you for this annotation, we corrected the sentence.
(2) line 208. Reference.
àWe inserted the Reference.
Gargiulo, L.; May, M.; Rivero, E.M.; Copsel, S.; Lamb, C.; Lydon, J.; Davio, C.; Lanari, C.; Lüthy, I.A.; Bruzzone, A. A Novel Effect of β-Adrenergic Receptor on Mammary Branching Morphogenesis and its Possible Implications in Breast Cancer. Journal of Mammary Gland Biology and Neoplasia 2017, 22, 43-57, doi:10.1007/s10911-017-9371-1.
(3) line 266. 0.1 μM-treated data were not provided in this study.
à We are sorry, thank you for your careful reading. We corrected the mistake.
The medium was changed after 4 hours to pure DMEM without any FCS. After 20 hours the cells were stimulated with 1 μM epinephrine (Sigma-Aldrich Chemie GmbH, Munich, Germany).
(4) Treated time periods were described differently in the figure legends and methods; line 140, 6 h; line 293, 48 h.
à Thank you for this correction. 48h is correct. We corrected the mistake.
References
- Evangelos, D.K.; Andreas, S.; Dido, V. Evidence for L-Dopa Decarboxylase Involvement in Cancer Cell Cytotoxicity Induced by Docetaxel and Mitoxantrone. Current Pharmaceutical Biotechnology 2018, 19, 1087-1096, doi:http://dx.doi.org.emedien.ub.uni-muenchen.de/10.2174/1389201019666181112103637.
- Luthy, I.A.; Bruzzone, A.; Pinero, C.P.; Castillo, L.F.; Chiesa, I.J.; Vazquez, S.M.; Sarappa, M.G. Adrenoceptors: Non Conventional Target for Breast Cancer? Current Medicinal Chemistry 2009, 16, 1850-1862, doi:http://dx.doi.org.emedien.ub.uni-muenchen.de/10.2174/092986709788186048.
- Tremmel, E.; Hofmann, S.; Kuhn, C.; Heidegger, H.; Heublein, S.; Hermelink, K.; Wuerstlein, R.; Harbeck, N.; Mayr, D.; Mahner, S., et al. Thyronamine regulation of TAAR1 expression in breast cancer cells and investigation of its influence on viability and migration. Breast Cancer (Dove Med Press) 2019, 11, 87-97, doi:10.2147/bctt.S178721.
- Epinephrine. In Drugs and Lactation Database (LactMed), National Library of Medicine (US): Bethesda (MD), 2006.
- Gratz, M.J.; Stavrou, S.; Kuhn, C.; Hofmann, S.; Hermelink, K.; Heidegger, H.; Hutter, S.; Mayr, D.; Mahner, S.; Jeschke, U., et al. Dopamine synthesis and dopamine receptor expression are disturbed in recurrent miscarriages. Endocr Connect 2018, 7, 727-738, doi:10.1530/ec-18-0126.
- Vattai, A.; Akyol, E.; Kuhn, C.; Hofmann, S.; Heidegger, H.; von Koch, F.; Hermelink, K.; Wuerstlein, R.; Harbeck, N.; Mayr, D., et al. Increased trace amine-associated receptor 1 (TAAR1) expression is associated with a positive survival rate in patients with breast cancer. J Cancer Res Clin Oncol 2017, 10.1007/s00432-017-2420-8, doi:10.1007/s00432-017-2420-8.
- Schröder, L.; Marahrens, P.; Koch, J.G.; Heidegger, H.; Vilsmeier, T.; Phan-Brehm, T.; Hofmann, S.; Mahner, S.; Jeschke, U.; Richter, D.U. Effects of green tea, matcha tea and their components epigallocatechin gallate and quercetin on MCF‑7 and MDA-MB-231 breast carcinoma cells. Oncology reports 2019, 41, 387-396, doi:10.3892/or.2018.6789.
- Cerione, R.A. The experiences of a biochemist in the evolving world of G protein-dependent signaling. Cell Signal 2018, 41, 2-8, doi:10.1016/j.cellsig.2017.02.016.
- Radde, B.N.; Ivanova, M.M.; Mai, H.X.; Salabei, J.K.; Hill, B.G.; Klinge, C.M. Bioenergetic differences between MCF-7 and T47D breast cancer cells and their regulation by oestradiol and tamoxifen. The Biochemical journal 2015, 465, 49-61, doi:10.1042/bj20131608.
- Barzegar, E.; Fouladdel, S.; Movahhed, T.K.; Atashpour, S.; Ghahremani, M.H.; Ostad, S.N.; Azizi, E. Effects of berberine on proliferation, cell cycle distribution and apoptosis of human breast cancer T47D and MCF7 cell lines. Iranian journal of basic medical sciences 2015, 18, 334-342.
- Jahanafrooz, Z.; Motameh, N.; Bakhshandeh, B. Comparative Evaluation of Silibinin Effects on Cell Cycling and Apoptosis in Human Breast Cancer MCF-7 and T47D Cell Lines. Asian Pacific journal of cancer prevention : APJCP 2016, 17, 2661-2665.
- Cui, B.; Luo, Y.; Tian, P.; Peng, F.; Lu, J.; Yang, Y.; Su, Q.; Liu, B.; Yu, J.; Luo, X., et al. Stress-induced epinephrine enhances lactate dehydrogenase A and promotes breast cancer stem-like cells. The Journal of clinical investigation 2019, 129, 1030-1046, doi:10.1172/jci121685.
- Ouyang, X.; Zhu, Z.; Yang, C.; Wang, L.; Ding, G.; Jiang, F. Epinephrine increases malignancy of breast cancer through p38 MAPK signaling pathway in depressive disorders. International journal of clinical and experimental pathology 2019, 12, 1932-1946.

Reviewer 3 Report
In the study presented in their manuscript, Tremmel Y et al conducted a study to analyze the influence of DDC expression in breast cancer cells in vivo and in vitro and to specify the interaction of G-protein coupled receptors with epinephrine in breast cancer.
Overall, the results of this study are well-presented, and the manuscript is well-written. However, this article includes some issues to be resolved and clarified. My concerns/comments are below.
Results
- Figure 1 – 3. The tumors with higher expression of DDC were associated with smaller tumor size and better OS. However, the result of grade demonstrated the opposite result meaning that tumors with higher expression of DDC were associated with more advanced grade. How do you interpret these results?
- Regarding Figure 3, the number of patients in each group, IRS < 2 or IRS ≥ 2, should be indicated in the figure or figure legends.
- Figure 4, it is better to stay in the same trend with other figures. It would be better if T47D comes to the right side of the figure as other figures.
- I could not find the data (figure) for Line 144 – 145. The authors need to add the figure or make a comment such as data not shown.
- For Figure 6, despite the authors are demonstrating the difference of DDC/ß-Actin between the control and treated group by using a bar plot, it would be helpful for the readers if the authors list the actual number of DDC/ß-Actin below the picture of the Western Blot membrane. It is hard to tell the difference between the two groups by looking only at the picture of Western Blot membrane and bar plots would be more convincing.
- The error bars need to be inserted in the bar plots (Figure 4 - 7).
Discussion
- In Line 206-208, the authors stated that estrogen levels seemed to be decreased after epinephrine stimulation and concluded that “Less estrogen leads to a reduction of the MCF7 cell viability and metabolism”. However, they never demonstrated or validated the downregulation of estrogen. The authors cannot conclude a statement with the only speculation.
- Similar to the issue above, the authors made a statement as “epinephrine leads to a downregulation of estrogen synthesis and an upregulation of Gi- PcR in MCF7 cells.”, in the current study, the authors never demonstrated the downregulation of estrogen synthesis.
Conclusion
- I would suggest reconsidering and rewriting the conclusion section. The conclusion must include the answers to the hypothesis or aim of this study which seems missing in the current conclusion. The future plan and the limitation of the study (first and the last sentence of the paragraph respectively) need to be moved to the discussion section.
Materials and Methods
- It would be easier for the readers to understand the patients’ characteristics with a table instead of writing in Materials and Methods (Line 225 - 233). Also, providing hormonal status (Estrogen, Progesterone, and HER2) would be informative for the readers.
References
- For reference #2 and #3, the authors need to cite the latest versions. The latest version would be 2018 for each citation (PMID: 30350310 and PMID: 30207593).
Author Response
Responses to the reviewers
L-DOPA-Decarboxylase (DDC) is a positive prognosticator for breast cancer patients and Epinephrine regulates breast cancer cell growth in vitro according to DDC regulation
Reviewer 3
Comments and Suggestions for Authors
In the study presented in their manuscript, Tremmel et al conducted a study to analyze the influence of DDC expression in breast cancer cells in vivo and in vitro and to specify the interaction of G-protein coupled receptors with epinephrine in breast cancer.
Overall, the results of this study are well-presented, and the manuscript is well-written. However, this article includes some issues to be resolved and clarified. My concerns/comments are below.
Thank you for your review.
Results
- Figure 1 – 3. The tumors with higher expression of DDC were associated with smaller tumor size and better OS. However, the result of grade demonstrated the opposite result meaning that tumors with higher expression of DDC were associated with more advanced grade. How do you interpret these results?
à Thank you for this remark. That is an interesting fact for sure. We do see a connection, undifferentiated tumors (G3) are more aggressive, but because of their high division rate the chemotherapies can take effect in all- day clinical experience.[1] Further, there is no direct correlation between tumor size and grading.[2] Therefore, it is possible that DDC is expressed in small tumor sizes and in more advanced gradings. Surely, further investigations should be carried out to specify this template more. We added a supplementary table 1 for a better overview on the tumor characteristics of our breast cancer samples.
- Regarding Figure 3, the number of patients in each group, IRS < 2 or IRS ≥ 2, should be indicated in the figure or figure legends.
à Thank you for this annotation. We added the number of patient and the following paragraph:
Via immunohistochemical analysis of 235 samples of breast cancer, a significant upregulation of DDC protein expression could be observed in primary breast cancer with a tumor size below pT2 (r= -0.147; p= 0.032) (Figure 1). Additionally, an upregulation of DDC expression could be analyzed via immunochemistry with higher tumor grading (G1-G2 p=0.013; G1-G3 p= 0.008)) (Figure 2). Overexpression of DDC protein expression (intensity of 2 or more, n= 109) in the primary tumor was associated with a significantly longer overall survival (p= 0.03) of patients in the cohort compared to patients without a DDC overexpression (n= 126) dependent on the size of the tumor (Figure 3).
Figure 3 Influence of DDC expression on OS of patients with early breast cancer. Kaplan-Meier curve showing the association between an increased DDC expression (Intensity of 2 or more, n=109; Intensity < 2, n= 126) and a longer overall survival (p= 0.03).
- Figure 4, it is better to stay in the same trend with other figures. It would be better if T47D comes to the right side of the figure as other figures.
à Thank you for your remark. We changed the arrangement:
Figure 4 Western Blot analysis of DDC protein expression in MCF7 and T47D cells after stimulation with Epinephrine. a) Bar chart of DDC expression in MCF7 cells after incubation with 10µM Epinephrine. Epinephrine led to an increase of DDC protein expression (p= 0.051) which was not significant. 1)-6) Western Blot membranes after incubation with ß-Actin and DDC antibodies (marked with grey box). 1) 3) 5): control samples. 2) 4) 6): stimulated samples with 10µM Epinephrine. b) Bar chart of DDC expression in T47D cells after incubation with 10µM Epinephrine. Epinephrine induced a significant downregulation of DDC protein expression (p= 0.038). 1)-6) show Western Blot membranes after incubation with ß-Actin and DDC antibodies (marked with a grey box). 1) 3) 5): control samples. 2) 4) 6): stimulated samples with 10µM Epinephrine.
- I could not find the data (figure) for Line 144 – 145. The authors need to add the figure or make a comment such as data not shown.
Line 176: In contrast, MCF7 cells showed a significant change in - protein expression. After an incubation with 10mM epinephrine for 6 hours, a significant upregulation of - protein expression was observed (p= 0.008) (Figure 6).
à You can find this results in the bar chart in figure 6 and 7. And additionally in supplementary figure 1.
- For Figure 6, despite the authors are demonstrating the difference of DDC/ß-Actin between the control and treated group by using a bar plot, it would be helpful for the readers if the authors list the actual number of DDC/ß-Actin below the picture of the Western Blot membrane. It is hard to tell the difference between the two groups by looking only at the picture of Western Blot membrane and bar plots would be more convincing.
à Figure 6 shows the analysis of - protein expression in MCF7 cells after stimulation with Epinephrine. See the figure legend below:
Figure 6 Western Blot analysis of - protein expression in MCF7 cells after stimulation with Epinephrine. Bar chart of - protein expression in MCF7 cells after incubation with 10µM Epinephrine. Epinephrine induced a significant upregulation of - protein expression (p= 0.008). 1)-6) Western Blot membranes after incubation with ß-Actin and - protein antibodies. 1) 3) 5): control samples. 2) 4) 6): stimulated samples with 10µM Epinephrine.
The Western Blot membrane itself below the bar chart shows for the antibody reaction. There are two different membranes, one for ß-Actin and one for - protein. We dispensed with an extra marker in the original picture. We validated each of our experiments nine times. The densitometry showed significant results for the DDC protein and the - Protein expression on protein level (validated through comparing to ß-Actin expression) and was validated nine times.
- The error bars need to be inserted in the bar plots (Figure 4 - 7).
à Thank you for your annotation. The bar charts show the mean of relative expression and not the raw data. Therefore, the presentation of error bars is not applicable in this case. [3] Further it is not allowed to add error bars on mean values of relative expression.
Discussion
- In Line 206-208, the authors stated that estrogen levels seemed to be decreased after epinephrine stimulation and concluded that “Less estrogen leads to a reduction of the MCF7 cell viability and metabolism”. However, they never demonstrated or validated the downregulation of estrogen. The authors cannot conclude a statement with the only speculation.
à In our discussion we discuss our results and the current literature. The consequence, that estrogen is relevant for the reduction of MCF7 cell viability is in line with the investigation of O’ Mahony et al. (2012), who focused on the impact of estradiol on the metabolism of glycolysis of MCF-7 cells and showed that in a setting of low extracellular glucose levels 17-estradiol rescues cell viability through upregulation of ATP and the activity of pyruvate dehydrogenase, which induces the tricarboxylic acid cycle and suppresses glycolysis. [4] The result is not a speculation. but a result, for which we found concordance in literature. We agree with you, that in future we have to further investigate on this topic.
We added the paragraph (line 270):
The level of estrogen seems to drop under stimulation with epinephrine. Less estrogen leads to a reduction of the MCF7 cell viability and metabolism. This finding is in line with the investigation of O’ Mahony et al. (2012), who focused on the impact of estradiol on the metabolism of glycolysis of MCF-7 cells and showed that in a setting of low extracellular glucose levels 17-estradiol rescues cell viability through upregulation of ATP and the activity of pyruvate dehydrogenase, which induces the tricarboxylic acid cycle and suppresses glycolysis. [4] Here further investigations are needed to specify in more detail.
- Similar to the issue above, the authors made a statement as “epinephrine leads to a downregulation of estrogen synthesis and an upregulation of Gi- PcR in MCF7 cells.”, in the current study, the authors never demonstrated the downregulation of estrogen synthesis.
à Thank you for your annotation. We corrected the paragraph in line 294, please see below:
This is in line with our research result, that epinephrine seems to induce a downregulation of estrogen synthesis and an upregulation of - PcR in MCF7 cells. This signaling pathway should be part of further investigations. The signaling latter known so far involves the cAMP-nuclear binding element (CREB), which is regulated through the cAMP level. [33] cAMP is a product of the adenylate cyclase, which is inhibited through - PcR.
Conclusion
I would suggest reconsidering and rewriting the conclusion section. The conclusion must include the answers to the hypothesis or aim of this study which seems missing in the current conclusion. The future plan and the limitation of the study (first and the last sentence of the paragraph respectively) need to be moved to the discussion section.
Thank you for your constructive annotation.
We added the following paragraph at the end of the discussion section (line 298):
Aims of future studies include further investigation of the functional interaction between -PcR and the DDC in breast cancer tissue and cells. Further studies are required to analyse the signalling pathways of the endogenous ligands and to possibly re-examine adrenergic signalling in breast cancer.
Rewritten conclusion section inserted in (line 410):
L-DOPA-Decarboxylase could be identified as one positive prognosticator for breast cancer patients with primary tumor. Epinephrine, as a substrate of the DDC, influences the viability of breast cancer cells in vitro. DDC may represent a novel target for the prevention and treatment of breast cancer.
Materials and Methods
It would be easier for the readers to understand the patients’ characteristics with a table instead of writing in Materials and Methods (Line 225 - 233). Also, providing hormonal status (Estrogen, Progesterone, and HER2) would be informative for the readers.
à This is a relevant annotation. But because of the huge number of patients (n=235) and the various characteristics breast cancer cells have, a table would overload the extent of this paper. Further the receptor status was specified in our laboratory, but we did not focus on this in our paper. The receptor status is not correlated to the size or histological grading; therefore, we did not focus on it in this paper. It is surely an interesting aspect for future investigations about this topic in sum with the exact signaling pathway including estrogen. In accordance to your annotation we added table 1 with the patient’s characteristics of the collective (line 316):
References
For reference #2 and #3, the authors need to cite the latest versions. The latest version would be 2018 for each citation (PMID: 30350310 and PMID: 30207593).
à The references have been updated to the latest versions.
- Ferlay, J.; Colombet, M.; Soerjomataram, I.; Mathers, C.; Parkin, D.M.; Piñeros, M.; Znaor, A.; Bray, F. Estimating the global cancer incidence and mortality in 2018: GLOBOCAN sources and methods. International journal of cancer 2019, 144, 1941-1953, doi:10.1002/ijc.31937.
- Bray, F.; Ferlay, J.; Soerjomataram, I.; Siegel, R.L.; Torre, L.A.; Jemal, A. Global cancer statistics 2018: GLOBOCAN estimates of incidence and mortality worldwide for 36 cancers in 185 countries. CA: a cancer journal for clinicians 2018, 68, 394-424, doi:10.3322/caac.21492.
References
- Harbeck, N.; Gnant, M. Breast cancer. Lancet 2017, 389, 1134-1150, doi:10.1016/S0140-6736(16)31891-8.
- Sopik, V.; Narod, S.A. The relationship between tumour size, nodal status and distant metastases: on the origins of breast cancer. Breast cancer research and treatment 2018, 170, 647-656, doi:10.1007/s10549-018-4796-9.
- Kolben TM, R.E., Vattai A, Hester A, Kuhn C, Schmoeckel E, Mahner S, Jeschke U, . PPARγ Expression Is Diminished in Macrophages of Recurrent Miscarriage Placentas.
- O'Mahony, F.; Razandi, M.; Pedram, A.; Harvey, B.J.; Levin, E.R. Estrogen modulates metabolic pathway adaptation to available glucose in breast cancer cells. Mol Endocrinol 2012, 26, 2058-2070, doi:10.1210/me.2012-1191.

Round 2
Reviewer 1 Report
The authors performed a huge improvement in the manuscript by following many of the recommendations. I add some suggestions about the revised document:
- Supplementary table 1 is hard to read, enlarge it and put a figure caption at the end instead of to the side
- Figure 3, indicate with letter or symbol (2 or more), unify criteria
- In the table of patients include the mean age (and range), as well as the state of menopause
- In the statistics section include the software used for the analysis
- Update abbreviations (for example TAAR)
- In the conclusions section, "further investigations/studies" sentence is indicated at least seven times which I think is too much redundant.
Author Response
Response to the reviewer- Minor Revision
L-DOPA-Decarboxylase (DDC) is a positive prognosticator for breast cancer patients and Epinephrine regulates breast cancer cell (MCF7 and T47D) growth in vitro according to their different expression of G_i- protein- coupled receptors
Reviewer 1
Thank you for your positive feedback and praise. We are glad to hear, that we could improve our paper.
We improved the following remarks. See below:
- Supplementary table 1 is hard to read, enlarge it and put a figure caption at the end instead of to the side.
à Thank you for your advice. We enlarged the supplementary table 1 and arranged it in three columns. Now all the data is available on one page. The figure caption is located at the end.
- Figure 3, indicate with letter or symbol (2 or more), unify criteria
à We indicated “2 or more” with symbol (≥) now in the figure 3 and the figure legend 3. It is unified.
Figure legend:
Figure 3 Influence of DDC expression on OS of patients with early breast cancer. Kaplan-Meier curve showing the association between an increased DDC expression (Intensity ≥2, n=109; Intensity < 2, n= 126) and a longer overall survival (p= 0.03).
- In the table of patients include the mean age (and range), as well as the state of menopause
à Thank you for this remark. We added the mean age in table 1. Unfortunately, the menopause status was not inquired in the patient’s data. The age could give an indication but is not a valid criteria for menopause status. We agree that this is an interesting characteristic and hope to include it for further studies. In case of the short time given for minor revision round, we are not able to include it now.
- In the statistics section include the software used for the analysis
à Thank you for this remark. The software is mentioned in the statistics section.
Line 413: Analysis of statistical data were transacted with IBM SPSS Statistics for Windows, Version 22.0. Armonk, NY: IBM Corp.
- Update abbreviations (for example TAAR)
à Thank you for this relevant annotation. We updated the abbreviations. See the table below or in the paper line 469.
MDPI |
Multidisciplinary Digital Publishing Institute |
DOAJ |
Directory of open access journals |
TLA |
Three letter acronym |
LD |
linear dichroism |
DDC |
L-Dopa Decarboxylase |
TAAR1 |
trace amine-associated receptor 1 |
OS |
overall survival |
TAs |
trace-amines |
AR |
adrenoreceptors |
PKA |
adenylcylase |
CREB |
cAMP-responsive element binding protein |
-PcR |
- protein coupled receptor |
ER |
estrogen receptor |
PR |
progesterone receptor |
- In the conclusions section, "further investigations/studies" sentence is indicated at least seven times which I think is too much redundant.
à Thank you for your remark. We reduced the number of sentences, which indicate that we are planning further studies.
Line 247: We removed the note of further investigations.
Line 298: We removed the sentence.
Line 313: Analyses of the signalling pathways of the endogenous ligands and to possibly re-examine adrenergic signalling in breast cancer need to be done.

Reviewer 2 Report
The authors adequately addressed my comments and suggestions.
Author Response
The authors adequately addressed my comments and suggestions.
Answer: Thank you very much!